# Oculo-Facio-Cardio-Dental Syndrome: A Case Report about a Rare Pathological Condition

**DOI:** 10.3390/ijerph16060928

**Published:** 2019-03-14

**Authors:** José Martinho, Hugo Ferreira, Siri Paulo, Anabela Paula, Carlos-Miguel Marto, Eunice Carrilho, Manuel Marques-Ferreira

**Affiliations:** Institute of Endodontics, Institute of Integrated Clinical Practice, Institute of Experimental Pathology, Institute for Clinical and Biomedical Research (iCBR), Area of Environment Genetics and Oncobiology (CIMAGO), CNC.IBILI, Faculty of Medicine, University of Coimbra, Avenida Bissaya Barreto, 3000-075 Coimbra, Portugal; hugidi.f@gmail.com (H.F.); sirivpaulo@gmail.com (S.P.); anabelabppaula@sapo.pt (A.P.); mig-marto@hotmail.com (C.-M.M.); eunicecarrilho@gmail.com (E.C.); m.mferreira@netcabo.pt (M.M.-F.)

**Keywords:** BCOR, dental anomalies, oculo-facio-cardio-dental syndrome, orthodontic treatment, orthognathic surgery, radiculomegaly

## Abstract

(1) Background: Oculo-facio-cardio-dental (OFCD) syndrome is a rare pathological condition with an X-linked dominant trait that only occurs in females; no males are born with OFCD syndrome. This syndrome is characterized by congenital cataracts with secondary glaucoma ocular defects, ventricular and atrial septal defects, or mitral valve prolapses. Facial traits are a long narrow face and a high nasal bridge with a bifid nasal tip. Dental anomalies include radiculomegaly, oligodontia, root dilacerations, malocclusion, and delayed eruption. (2) Methods: This clinical report describes a 26-year-old girl who suffers from OFCD syndrome and who was treated with a multidisciplinary approach. The treatment plan included orthodontic treatment, orthognathic surgery, namely LeFort I and a Bilateral Sagittal Split Osteotomy, and occlusal rehabilitation with implants. (3) Discussion: Early diagnosis and multidisciplinary treatment of orthodontic, orthognathic surgery and occlusal rehabilitation with implants make it possible to maintain tooth function and improve aesthetics with good prognoses for success. In this paper, we report a case of a female patient with OFCD syndrome, who was referred for orthodontic treatment and occlusal rehabilitation and treated with a multidisciplinary approach.

## 1. Introduction

Oculo-facio-cardio-dental (OFCD) syndrome is defined as a rare pathological condition first introduced in 1996 by Gorlin et al. [1]. Oculo-facio-cardio-dental syndrome only occurs in females and no ma Fles are born with OFCD syndrome. The typical facial features of OFCD are ocular defects including microphthalmia, congenital cataract, a long narrow face, and a high nasal bridge with separation of the anterior cartilages. The most common dental abnormalities are radiculomegaly, oligodontia, retained primary teeth, and malocclusion [1,2]. Just two years later, the same authors reported another case characterized by root dilacerations of four canines, malocclusion, and cataracts [3,4].

Genetic analysis of OFCD syndrome revealed that it is caused by a mutation in the encoding BCL-6 interacting corepressor (BCOR), an X-linked dominant trait presumed to be responsible for lethality in affected males [1,5,6,7,8,9]. The BCOR gene provides information for the protein and corepressor, which seems to play an imperative function in the first steps of embryogenic growth. Truncating, frameshift mutations, deletions, and nonsense mutations in the gene result in a premature termination at chromosome Xp114 [10]. Early diagnosis of this syndrome is crucial to establish a correct treatment plan to obtain the best results. Diagnosis can be made after the age of 15 by a multidisciplinary team, and the differential diagnosis can be made with rubella and rubeola [11,12].

In this paper, we report a case of a patient with OFCD syndrome who was referred for orthodontic treatment and occlusal rehabilitation with treatment by a multidisciplinary team.

## 2. Case Summary

A 26-year-old girl was referred in 2002 by her orthognathic surgeon for orthodontic treatment and oral rehabilitation because she did not like her face, gummy smile, and crowded teeth. She wanted straight teeth, a rounder face, and a less pronounced chin. The patient had Class II malocclusion on a Class III skeletal base with a prognathic mandible, increased facial proportions, and facial asymmetry. Intraorally, there was localized crowding in the upper and lower arches, a nine mm deep bite, an anterior and bilateral scissors bite with signs of functional disturbances, and occlusal trauma.

The clinical examination revealed a misalignment of her thumbs, valgus foot condition, and a long, narrow face (dolichocephalous) with a bifid tip of the nose and very deep-set eyes with cataracts (Figure 1). The patient had a prolapsed mitral valve.

Clinical oral examination revealed dentition with numerous missing teeth (Table 1 and Figure 2), and the panoramic radiograph showed radiculomegaly in the upper right central incisor, upper left canine, first lower left premolar, lower right and left canine, lower right central incisor, lower left central incisor, lower right lateral incisor, and first lower right and left premolar.

The first upper left molar, upper right canine, upper left lateral incisor, first upper left premolar, first upper right molar, first lower right molar, and first lower left molar are absent (Figure 3).

A lateral cephalogram performed before orthodontic treatment to analyze linear and angular measurements showed significant craniofacial dysmorphia (Figure 4A).

## 3. Results—Treatment Sequence

Taking this malocclusion into account, the multidisciplinary team confirmed that orthodontic treatment alone was insufficient to adequately correct this functional case. The treatment plan consisted of orthodontic treatment and orthognathic surgery to solve dental malposition and skeletal anomalies, and oral rehabilitation with implants was planned in the second step (Table 2).

To begin the treatment, an upper mouthguard was placed to disclose the teeth and open the bite, and lower brackets were bonded with a 0.016″ NiTi archwire (Figure 5). Six months later, the upper bracket was directly bonded, and a 0.016″ NiTi archwire was placed. During the patient’s orthodontic treatment, light forces were applied to minimize the risk of ankylosis (Figure 6A).

After a control orthopantomography, a radiolucency and asymptomatic lesion appeared in the left mandibular branch. To guide the surgeon in proper patient management, a computer tomography was done. The lesion was not related to the mandibular molar tooth and was surgically removed in the hospital. After three months, the patient was subjected to a LeFort I surgery to make a 5 mm superior reposition of the maxilla to reduce the gummy smile. At the same time, a bilateral sagittal split osteotomy (BSSO) was made for mandibular setback and reposition. A genioplasty for vertical reduction with 6 mm was also performed. After osteotomy, small bone plates with bone screws were used to stabilize the segments and to obtain a rigid fixation (Figure 4B and Figure 6B).

The orthodontic appliance was removed after five years and retainers with missing teeth were made to provide retention. Oral rehabilitation was performed one year after removal of the appliance with implants: A Ø3.5 mm × 10.5 mm (Southern Implants) implant was placed in the upper right lateral incisor region, a Ø3.5 mm × 10 mm (Nobel Replace) implant was placed in the upper left canine region, and a Ø4 mm × 13 mm (Neodent) implant was placed in the first upper left premolar region.

Peri-implantitis occurred along the first upper left premolar implant one year after placement. Curettage peri-implantitis and prescription of amoxicillin and metronidazole every 8 h for 14 days were carried out, but after six months the implant was removed. At a follow-up of the treatment after four years and six months, the patient was stable and happy with the results (Figure 7, Figure 8 and Figure 9).

## 4. Discussion

The association of facial, ocular, and dental features are the most important features for the diagnosis of OFCD [8,11,12]. Dentists and orthodontists might be in the best position to diagnose this syndrome because they work with dental radiographs. They can therefore compare the eruption, root formation, and apical closure of teeth throughout the orthodontic treatment period. The OFCD syndrome is very uncommon, and no more than 100 cases have been reported [13]. The features of this syndrome present a great variety of signs such as dental anomalies that are difficult to detect at birth. It is important to make a differential diagnosis between OFCD and Marfan syndrome. Marfan syndrome encompasses not only the oculo-facio-cardio-dental anomalies, but also the musculoskeletal, central nervous, pulmonary, and integumentary systems. In this case report, the patient presented clinical characteristics similar to those of Marfan syndrome [14,15,16]. However, the presence of mandibular prognathism, the bifid tip of the nose, and several teeth with radiculomegaly are pathognomonic signals of OFCD syndrome.

In this clinical case, we reported several dental abnormalities such as canine radiculomegaly associated with congenital cataracts and other facial treatments which are typical of OFCD syndrome.

Taking into account the Index of Orthognathic Functional Treatment Need (IOFTN) ([17,18,19], skeletal anomalies (5.7), an anterior and bilateral scissors bite with signs of functional disturbances and occlusal trauma (5.5), a deep bite with dental and soft tissue trauma (4.8), and gingival exposure ≥ 3 mm at rest (4.9) confirmed, for the multidisciplinary team, that orthodontic treatment alone was insufficient to adequately correct this functional case. In this case report, the patient had a very bad malocclusion which made it difficult to install the orthodontic appliances. For this reason, we began the treatment with an upper mouth guard to permit the lower orthodontic appliance to be installed.

After orthodontic treatment and orthognathic surgery, the malpositioned permanent teeth bimaxillary skeletal discrepancy improved, and edentulous space was restored with implants to establish better occlusal harmony. The implant put in the first upper left premolar region was lost due to peri-implantitis. This may be related to OFCD syndrome and a radiopacity seen in the orthopantomography image, or the lack of blood in bone irrigation in this edentulous space. For rehabilitation of the edentulous space due to the loss of the implant, the treatment plan was changed and in agreement with the patient, it was decided to make a bridge. The duration of this treatment was about five years. Besides being a clinical case that included orthodontic and surgical treatment, it was necessary to adapt the different phases of the plan to the dental characteristics, namely the excessive size of the roots. Due to the appearance of setbacks during treatment, namely the loss of the above-mentioned implant and the appearance of a lesion of asymptomatic apical periodontitis associated with tooth 35, the treatment plan was extended.

Although the patient was subjected to a LeFort I surgery to make a 5 mm superior reposition of the maxilla, the patient still presents a somewhat gummy smile, but she is pleased and has no complaints. Our case report presented an adult patient with many of the signs described in the literature that could confirm the pathognomonic signals which are characteristic of OFCD syndrome. Based on the clinical and radiographic findings, the therapeutic approach to this case was performed by a multidisciplinary team that included orthognathic surgery, orthodontic correction, and occlusal rehabilitation.

In order to complete this case report, it would be important to do a BCOR (BCL-6 interacting corepressor) mutation analysis, which could not be performed due to the patient’s state of health.

## 5. Conclusions

These data show that early diagnosis, prompt treatment, and a multidisciplinary approach are a suitable method to solve the skeletal, facial, and dental disharmony in patients with oculo-facio-cardio-dental syndrome.

## Figures and Tables

**Figure 1 ijerph-16-00928-f001:**
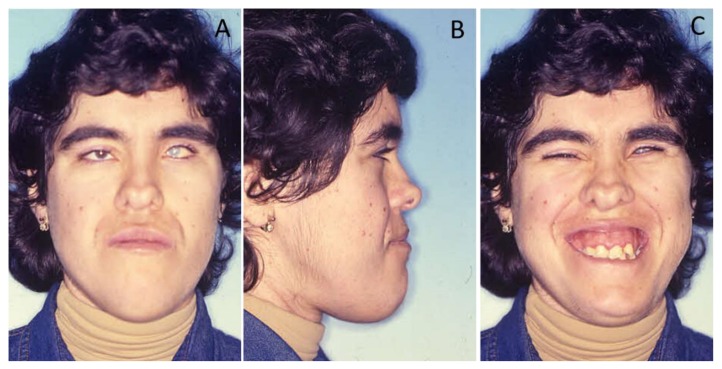
Pre-treatment extra-oral photographs of the oculo-facio-cardio-dental (OFCD) syndrome patient showing facial and dental characteristics of the OFCD syndrome in frontal (**A**), profile (**B**), and smile (**C**) photographs.

**Figure 2 ijerph-16-00928-f002:**
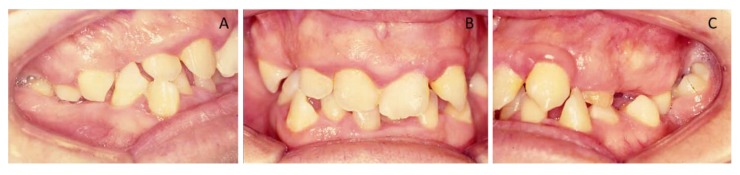
Pre-treatment, intra-oral photographs of patient showing right (**A**), frontal (**B**), and left occlusion (**C**). Note the missing teeth, gummy smile, and severe malocclusion.

**Figure 3 ijerph-16-00928-f003:**
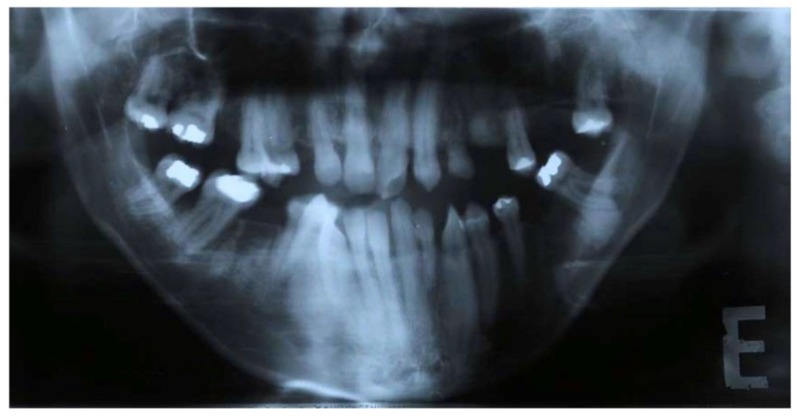
Pre-treatment panoramic radiograph of patient showing several missing teeth, radiculomegaly, and dilaceration of roots.

**Figure 4 ijerph-16-00928-f004:**
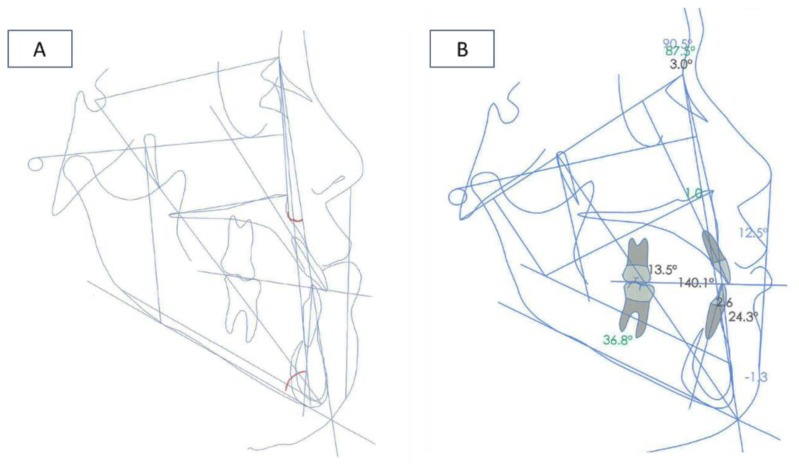
Cephalogram. (**A**) Pre-surgery and (**B**) post-surgery.

**Figure 5 ijerph-16-00928-f005:**
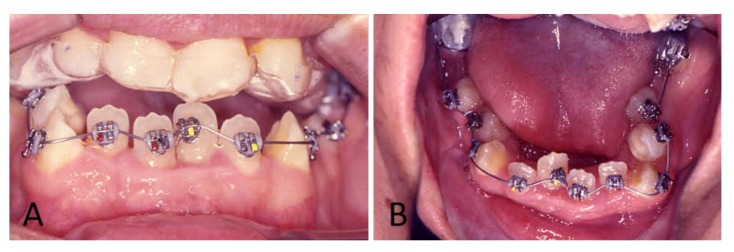
To start the treatment, an upper mouthguard (**A**) and lower bracket direct bonding and a 0.016” NiTi archwire (**B**) were placed.

**Figure 6 ijerph-16-00928-f006:**
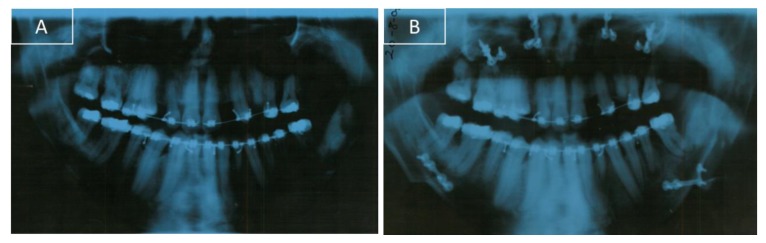
Orthopantomography during orthodontic treatment. (**A**) Before surgery with the fixed appliances. (**B**) After surgery.

**Figure 7 ijerph-16-00928-f007:**
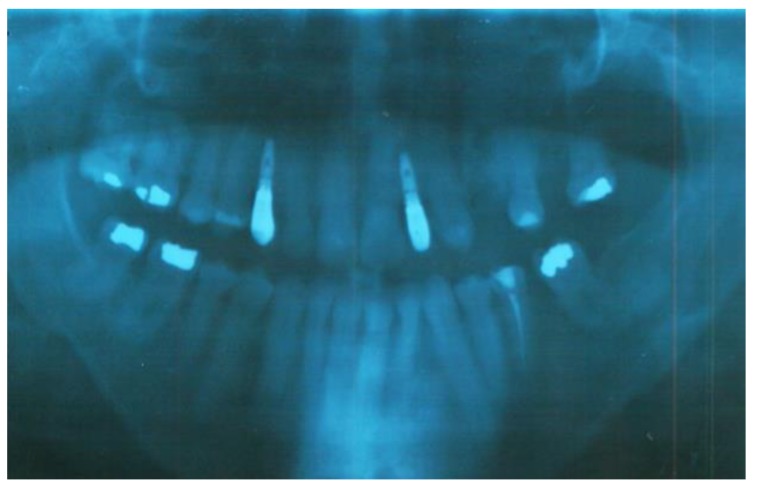
Final orthopantomography.

**Figure 8 ijerph-16-00928-f008:**
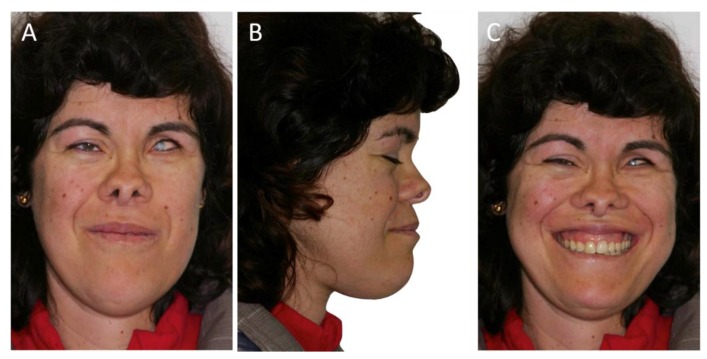
Extra-oral photographs after treatment (**A**–**C**). At four years and six months, she showed a less severe gummy smile and a more symmetric face.

**Figure 9 ijerph-16-00928-f009:**
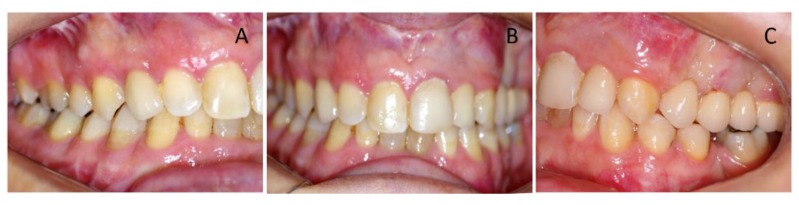
Post-treatment intra-oral photographs (**A**–**C**). At four years and six months, they showed a stable occlusion with orthodontic treatment, orthognathic surgery, and oral rehabilitation.

**Table 1 ijerph-16-00928-t001:** Table of the dentition showing which teeth are missing and have anomalies.

**18**	**17**	**16**	**15**	**14**	**13**	**12**	**11**	**21**	**22**	**23**	**24**	**25**	**26**	**27**	**28**
N	N	A	A	N	A	N	N	R	A	R	A	N	A	A	N
**48**	**47**	**46**	**45**	**44**	**43**	**42**	**41**	**31**	**32**	**33**	**34**	**35**	**36**	**37**	**38**
N	N	A	N	R	R	R	R	R	N	R	R	N	A	A	N

N (normal); A (absent); R (tooth with radiculomegaly).

**Table 2 ijerph-16-00928-t002:** Treatment plan of this case report.

Treatment Phase	Treatment	Objective
Phase 1	Upper mouthguard Lower fixed appliance	Disocclusion and possibility of placement of upper fixed appliance.
Phase 2	Upper fixed appliance	Preparation of the occlusion for the surgical phase
Phase 3	Orthognathic surgery—LeFort I with 5 mm maxillary impaction	Gummy smile reduction
	Bilateral sagittal split osteotomy	Mandibular setback reposition
	Genioplasty with 6 mm of vertical reduction	Improve the facial aesthetic with correction of asymmetry
Phase 4	Oral Rehabilitation	Prosthetic rehabilitation of missing teeth

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
