# Peer review of "Oculo-Facio-Cardio-Dental Syndrome: A Case Report about a Rare Pathological Condition"

_ijerph, 2019, doi:10.3390/ijerph16060928_

Round 1

Reviewer 1 Report

Thank you for submitting this case report.  It was very interesting.

I have a few comments and suggestions

Introduction:

It would be helpful to expand on the known genetics of this syndrome.  Also to define BCOR.

In line 60-61 you say the OFCD is caused by a mutation in BCOR and a X-linked dominant trait or are you saying the BCOR is an X-linked dominant trait.

Methods:

You mention in line 82 numerous missing teeth.  you could mention that their are retained primary teeth and missing permanent teeth. 

The information listed in lines 82-92 might fit nicely in a graphic or table of the dentition showing which teeth are missing and have anomalies. 

Why were different implant types/brands placed?  Why not one system?

Results:

Peri-implantitis - please be consistent in how you use the term, line 115 and 116

For the peri-implantitis, was a flap reflected to evaluate the implant? bone grafting,etc.  You could elaborate in the discussion if you thought this had anything to do with OFCD or possibly due to poor blood supply after her LeFort I surgery?

You mention in line 117 that the implant for the upper right 1st PM was removed after 6months, but in the photos the patient has this tooth.  Was another implant placed, and if another implant was not placed, why - this could be touched on in the discussion.

For the lower left molar that had a cyst associated with it, was the tooth removed as well?

Elaborate on why you made an upper mouthguard and delay maxillary bracket placement. It is assumed that the reader will know. 

The case report is about a rare syndrome, OFCD, but is also about how you treatment planned and restored this patient.  In the results section, it might be useful to elaborate on the treatment plan process and why you chose to do certain treatments and how did the patient's input change your treatment plan/duration if at all.  Try not to assume that the reader will know why you did something a particular way or what you rationale was. I think this will help strengthen your case report and provide more depth. 

Author Response

Point 1: Introduction: It would be helpful to expand on the known genetics of this syndrome. Also to define BCOR.

Response 1: Alterations were made in the manuscript.

Point 2: In line 60-61 you say the OFCD is caused by a mutation in BCOR and a Xlinked dominant trait or are you saying the BCOR is an X-linked dominant trait.

Response 2: Alterations were made in the manuscript.

Point 3: Methods: You mention in line 82 numerous missing teeth.  you could mention that their are retained primary teeth and missing permanent teeth. The information listed in lines 82-92 might fit nicely in a graphic or table of the dentition showing which teeth are missing and have anomalies.

Response 3: Table of the dentition showing which teeth are missing and have anomalies was made in the manuscript.

Point 4: Why were different implant types/brands placed?  Why not one system?

Response 4: Several implant systems were used due to change of the system used in the clinic.

Point 5: Results: Peri-implantitis - please be consistent in how you use the term, line 115 and 116

Response 5: Alterations were made in the manuscript.

Point 6: For the peri-implantitis, was a flap reflected to evaluate the implant? bone grafting,etc.  You could elaborate in the discussion if you thought this had anything to do with OFCD or possibly due to poor blood supply after her LeFort I surgery?

Response 6: Alterations were made in the manuscript.

Point 7: You mention in line 117 that the implant for the upper right 1st PM was removed after 6months, but in the photos the patient has this tooth.  Was another implant placed, and if another implant was not placed, why - this could be touched on in the discussion.

Response 7: Alterations were made in the manuscript, explaining the change of treatment plan.

Point 8: For the lower left molar that had a cyst associated with it, was the tooth removed as well?

Response 8: Alterations were made in the manuscript, explaining why the tooth wasn’t removed.

Point 9: Elaborate on why you made an upper mouthguard and delay maxillary bracket placement. It is assumed that the reader will know. The case report is about a rare syndrome, OFCD, but is also about how you treatment planned and restored this patient.

Response 9: Alterations were made in the manuscript, justifying the delay maxillary bracket placement.

Point 10: In the results section, it might be useful to elaborate on the treatment plan process and why you chose to do certain treatments and how did the patient's input change your treatment plan/duration if at all.  Try not to assume that the reader will know why you did something a particular way or what you rationale was. I think this will help strengthen your case report and provide more depth.

Response 10: Alterations were made in the manuscript. It was made a treatment plan in the results section, and it was discussed later.

The English language and style were reviewed by an English native throughout the article content. The proofreading declaration is sent as an attachment to the editor.

Reviewer 2 Report

Better specify the direction and amount of all surgical pieces (not only the vertical movement of the maxilla).

Add to the figures the pre and postsurgical radiographs and cephalometric tracings.

Author Response

Point 1: Better specify the direction and amount of all surgical pieces (not only the vertical movement of the maxilla). Add to the figures the pre and postsurgical radiographs and cephalometric tracings.

Response 1: Alterations were made in the manuscript. Cephalometric and orthopantomography images were added as suggested by the reviewer.

The English language and style were reviewed by an English native throughout the article content. The proofreading declaration is sent as an attachment to the editor.

Reviewer 3 Report

the paper needs a great deal of revision for grammar and style, there are many typos

abstract and introduction, please mention that Oculo-facial-cardio-dental syndrome only occurs in females and no males are born with OFCD syndrome.

line 34, correct to'X-linked dominant trait'

use capital "C" in Class III

use a systematic approach to describe the case,

she presented with a Class II malocclusion on a Class III Skeletal base with prognathic mandible, increased facial proportions and facial asymmetry. Intraorally there was deepbite (was there traumatic deepbite?), anterior and posterior x-bites, localised crowding in upper and lower arches

do you have better intraoral photos? and a better panoramic view as it does not show the details

you mentioned she had missing teeth, were they congenitally missing or have been extracted?

what is 'radiculomegaly' did you mean macrodontia? larger than average teeth?

you have mentioned 'The first upper left molar, upper left canine, upper right lateral incisor, first upper right premolar, first upper right molar, first lower right molar and first lower left molar are absent' I think this is not correct and based on your panoramic view and intraoral views it should be 'The upper right 1st molar, upper right canine, upper left lateral incisor, upper left first premolar, upper left 1st and 2nd molar, and both lower 1st molars were absent'

change the heading of 'results' to 'treatment sequence',

line 97, change 'were put' to 'we put', same line', change lower bracket direct bonding' to 'lower bracket were directly bonded' or 'lower brackets were bonded'

line 101, 102, what do you mean by 'For suspected cyst in the mandible branch left, a Computer Tomography was made, which was surgically removed.' give the exact location of the cyst, what type of cyst was it?

next line 'After 3 months, the patient was subjected to a LeFort I surgery for 5 mm maxillary impaction, BSSO (Bilateral Sagittal Split Osteotomy) for retrognathic reposition and genioplasty for vertical reduction.' correct to' After 3 months, the patient was subjected to a LeFort I osteotomy and patient had 5 mm maxillary impaction, mandibular setback (Bilateral Sagittal Split Osteotomy) to retroposition mandible as well as reduction genioplasty (vertical)'. did you do any maxillary advancement as well?

figure 3, what do you mean by the delayed eruption, which tooth, it is not visible on x-ray, you need a better x-ray(panoramic)

line 111 correct 'to retention' to 'provide retention'

line 118, correct 'is stable' to 'was stable'

add a panoramic view after orthognathic surgery and implant placement

add a cephalogram before and after surgery

read this section and improve your literature review

https://ghr.nlm.nih.gov/condition/oculofaciocardiodental-syndrome

Author Response

Comments and Suggestions for Authors

Point 1: the paper needs a great deal of revision for grammar and style, there are many typos

Response 1: The English language and style were reviewed by an English native throughout the article content. The proofreading declaration is sent as an attachment to the editor.

Point 2: abstract and introduction, please mention that Oculo-facial-cardio-dental syndrome only occurs in females and no males are born with OFCD syndrome.

Response 2: Alterations were made in the manuscript.

Point 3: line 34, correct to'X-linked dominant trait'

Response 3: Alterations were made in the manuscript.

Point 4: use capital "C" in Class III

Response 4: Alterations were made in the manuscript.

Point 5: use a systematic approach to describe the case, 

she presented with a Class II malocclusion on a Class III Skeletal base with prognathic mandible, increased facial proportions and facial asymmetry. Intraorally there was deepbite (was there traumatic deepbite?), anterior and posterior x-bites, localised crowding in upper and lower arches

Response 5: Alterations were made in the manuscript.

Point 6: do you have better intraoral photos? and a better panoramic view as it does not show the details

Response 6: Alterations were made in the manuscript. The orthopantomography of several stages of the treatment and the final one were placed in the article.

Point 7: you mentioned she had missing teeth, were they congenitally missing or have been extracted?

Response 7: It was not possible to assess the cause of loss of missing teeth.

Point 8: what is 'radiculomegaly' did you mean macrodontia? larger than average teeth?

Response 8: Radiculomegaly means larger and longer roots than the average of the other teeth.

Point 9: you have mentioned 'The first upper left molar, upper left canine, upper right lateral incisor, first upper right premolar, first upper right molar, first lower right molar and first lower left molar are absent' I think this is not correct and based on your panoramic view and intraoral views it should be 'The upper right 1st molar, upper right canine, upper left lateral incisor, upper left first premolar, upper left 1st and 2nd molar, and both lower 1st molars were absent'

Response 9: Alterations were made in the manuscript

Point 10: change the heading of 'results' to 'treatment sequence', 

Response 10: Alterations were made in the manuscript

Point 11: line 97, change 'were put' to 'we put', same line', change lower bracket direct bonding' to 'lower bracket were directly bonded' or 'lower brackets were bonded'

Response 11: Alterations were made in the manuscript

Point 12: line 101, 102, what do you mean by 'For suspected cyst in the mandible branch left, a Computer Tomography was made, which was surgically removed.' give the exact location of the cyst, what type of cyst was it?

Response 12: Alterations were made in the manuscript. We do not know the type of cyst because the surgical remove was performed in the hospital, and we do not have access to the hospital data.

Point 13: next line 'After 3 months, the patient was subjected to a LeFort I surgery for 5 mm maxillary impaction, BSSO (Bilateral Sagittal Split Osteotomy) for retrognathic reposition and genioplasty for vertical reduction.' correct to' After 3 months, the patient was subjected to a LeFort I osteotomy and patient had 5 mm maxillary impaction, mandibular setback (Bilateral Sagittal Split Osteotomy) to retroposition mandible as well as reduction genioplasty (vertical)'. did you do any maxillary advancement as well?

Response 13: Alterations were made in the manuscript. It was performed a LeFort I surgery to make a 5 mm superior reposition of the maxilla to reduce the gummy smile, without any advancement. A bilateral sagittal split osteotomy for mandibular setback and reposition was, also, performed.

Point 14: figure 3, what do you mean by the delayed eruption, which tooth, it is not visible on x-ray, you need a better x-ray(panoramic)

Response 14: Alterations were made in the manuscript

Point 15: line 111 correct 'to retention' to 'provide retention'

Response 15: Alterations were made in the manuscript

Point 16: line 118, correct 'is stable' to 'was stable'

Point 16: Alterations were made in the manuscript

Point 17: add a panoramic view after orthognathic surgery and implant placement

add a cephalogram before and after surgery

Response 17: Alterations were made in the manuscript

Point 18: read this section and improve your literature review

https://ghr.nlm.nih.gov/condition/oculofaciocardiodental-syndrome

Response 18: Alterations were made in the manuscript

Reviewer 4 Report

Dear Authors the paper is good and it gives an interesting case with a rare disease. In literature it is possible to find some papers describing associations between cranio-maxillo-facial disorders and cardiac pathology, such as Marfan syndrome, but with no-syndromic subjects too. I'm sending you attached three papers about this topic, it could be useful for Authors to read and insert them in the manuscript in order to make it longer.

Author Response

R: Changes were made in the discussion according to the suggestions of the reviewer and considering the suggested articles.

The English language and style were reviewed by an English native throughout the article content. The proofreading declaration is sent as an attachment to the editor.

Round 2

Reviewer 1 Report

Thank you for your changes.  I think they add to the quality of the article and explains minor deficiencies. 

Author Response

The manuscript was reviewed again by an English speaker, who is a native and reviewer of school books and other scholarly works in English and who has been the director for 25 years of an English language school that teaches English in Coimbra, the International School. Attached we sent the Proofreading Declaration by the native English reviewer, Dave Tucker.

Reviewer 3 Report

Thank you for the revisions, 

paper still needs revision for grammar and. Style, 

Remove the method heading just add case summary 

a good addition would be the addition of IOFTN index score for this case with appears to be 4.8( deep Bite) you need to cite the relevant references for this index ( J Orthod. 2014 Jun;41(2):77-83.;J Plast Reconstr Aesthet Surg. 2016 Jun;69(6):796-801; Int J Pediatr Otorhinolaryngol. 2015 Jul;79(7):1063-6)

Add the dimension of the implant used ? Any bone grafting? 

In the discussion mention That there is still increased gingival exposure ( gummy smile) 

Author Response

Point 1: paper still needs revision for grammar and. Style, 

Response 1: The manuscript was again reviewed by an English speaker, who is a native and reviewer of school books and other scholarly works in English and who has been the director for 25 years of an English language school that teaches English in Coimbra, the International School. Attached we sent the Proofreading Declaration by the native English reviewer, Dave Tucker.

Point 2: Remove the method heading just add case summary 

Response 2: Alterations were made in the manuscript

Point 3: a good addition would be the addition of IOFTN index score for this case with appears to be 4.8( deep Bite) you need to cite the relevant references for this index ( J Orthod. 2014 Jun;41(2):77-83.;J Plast Reconstr Aesthet Surg. 2016 Jun;69(6):796-801; Int J Pediatr Otorhinolaryngol. 2015 Jul;79(7):1063-6)

Response 3: Alterations were made in the manuscript

Point 4: Add the dimension of the implant used ? Any bone grafting? 

Response 4: Alterations were made in the manuscript, namely the dimension of the implants. No bone grafting was done.

Point 5: In the discussion mention That there is still increased gingival exposure ( gummy smile) 

Response 5: Alterations were made in the manuscript.
